# Metabolomics-Based Screening of Inborn Errors of Metabolism: Enhancing Clinical Application with a Robust Computational Pipeline

**DOI:** 10.3390/metabo11090568

**Published:** 2021-08-26

**Authors:** Brechtje Hoegen, Alan Zammit, Albert Gerritsen, Udo F. H. Engelke, Steven Castelein, Maartje van de Vorst, Leo A. J. Kluijtmans, Marleen C. D. G. Huigen, Ron A. Wevers, Alain J. van Gool, Christian Gilissen, Karlien L. M. Coene, Purva Kulkarni

**Affiliations:** 1Department of Human Genetics, Radboud University Medical Center, 6525 GA Nijmegen, The Netherlands; Brechtje.Hoegen@radboudumc.nl (B.H.); Alan.Zammit@radboudumc.nl (A.Z.); Albert.Gerritsen@radboudumc.nl (A.G.); Steven.Castelein@radboudumc.nl (S.C.); Maartje.vandeVorst@radboudumc.nl (M.v.d.V.); Christian.Gilissen@radboudumc.nl (C.G.); 2Radboud Institute of Molecular Life Sciences, Radboud University Medical Center, 6525 GA Nijmegen, The Netherlands; Udo.Engelke@radboudumc.nl (U.F.H.E.); Ron.Wevers@radboudumc.nl (R.A.W.); Alain.vanGool@radboudumc.nl (A.J.v.G.); karlien.coene@radboudumc.nl (K.L.M.C.); 3Translational Metabolic Laboratory, Department of Laboratory Medicine, Radboud University Medical Center, 6525 GA Nijmegen, The Netherlands; Leo.Kluijtmans@radboudumc.nl (L.A.J.K.); Marleen.Huigen@radboudumc.nl (M.C.D.G.H.)

**Keywords:** untargeted metabolomics, next-generation metabolic screening, inherited metabolic diseases, data analysis, mass spectrometry, bioinformatics pipeline, clinical application, biomarkers

## Abstract

Inborn errors of metabolism (IEM) are inherited conditions caused by genetic defects in enzymes or cofactors. These defects result in a specific metabolic fingerprint in patient body fluids, showing accumulation of substrate or lack of an end-product of the defective enzymatic step. Untargeted metabolomics has evolved as a high throughput methodology offering a comprehensive readout of this metabolic fingerprint. This makes it a promising tool for diagnostic screening of IEM patients. However, the size and complexity of metabolomics data have posed a challenge in translating this avalanche of information into knowledge, particularly for clinical application. We have previously established next-generation metabolic screening (NGMS) as a metabolomics-based diagnostic tool for analyzing plasma of individual IEM-suspected patients. To fully exploit the clinical potential of NGMS, we present a computational pipeline to streamline the analysis of untargeted metabolomics data. This pipeline allows for time-efficient and reproducible data analysis, compatible with ISO:15189 accredited clinical diagnostics. The pipeline implements a combination of tools embedded in a workflow environment for large-scale clinical metabolomics data analysis. The accompanying graphical user interface aids end-users from a diagnostic laboratory for efficient data interpretation and reporting. We also demonstrate the application of this pipeline with a case study and discuss future prospects.

## 1. Introduction

Inborn errors of metabolism (IEMs) are genetically determined biochemical disorders that have severe clinical consequences, which mostly present at neonatal or childhood age, but also milder presentations are known in adult patients. These disorders are caused by the dysfunction of enzymes or cofactors, leading to disruption of a given biochemical pathway and accumulation of toxic compounds and/or abnormal energy metabolism. If left undiagnosed and untreated, IEMs can result in irreversible intellectual and physical disability, neurological damage and can even be fatal. Early detection and accurate diagnosis are crucial to be able to initiate personalized therapy as soon as possible to achieve optimal patient outcome. A limited selection of treatable IEMs with a relatively high incidence is covered in newborn screening [1]. For those IEMs not covered in newborn screening, biochemical diagnostics are offered at metabolic laboratories, which traditionally involves a selection of targeted metabolite assays based on the clinical phenotype of the patient. Although this approach has proven itself useful over the years, it faces the problem of false-negative diagnosis when the clinical phenotype is incomplete. Additionally, yet unknown metabolic deviations cannot be detected [2]. This issue can be overcome by performing unbiased metabolic profiling to provide a true metabolic fingerprint of an individual suspected to have an IEM.

High-resolution mass spectrometry (MS)-based untargeted metabolomics is being increasingly used for metabolic profiling [3,4,5,6]. Metabolomics is defined as the systematic measurement and analysis of small molecules, called metabolites, present in biological samples [7]. In contrast to targeted metabolomics, where only a limited panel of known analytes is measured, untargeted metabolomics is holistic in nature and aims to measure as many metabolites as possible in a biological sample, generating a metabolic fingerprint representative of a biochemical phenotype. The broad metabolome coverage offered by untargeted metabolomics offers an unprecedented opportunity for diagnostic screening of individual patients suspected of an IEM. This approach also holds great promise for biomarker discovery for IEMs as well as the identification of novel metabolic disorders [8,9]. We recently demonstrated the application of untargeted metabolomics for diagnostic screening for IEM, an approach we termed next-generation metabolic screening (NGMS) [10]. This approach uses ultra-high-performance liquid chromatography quadrupole time-of-flight mass spectrometry (UHPLC-QTOF-MS) for holistic metabolic profiling in the plasma of individual IEM-suspected patients. The NGMS approach was thoroughly validated for use in ISO:15189-accredited diagnostics, including testing of plasma samples from patients with 46 distinct IEMs [10]. From July 2020 on, NGMS has been used as a first-tier test for diagnostic screening for IEMs in our laboratory, and over 600 diagnostic reports have been generated. 

As is typical with untargeted metabolomics, this approach generates large volumes of data, and tens of thousands of mass features are detected at a high level of sensitivity and resolution, raising the need for computational tools that can help in processing and interpreting this data. A growing number of software tools have been developed for metabolomics data processing and analysis that are widely accepted and used in the scientific community [11,12,13]. Most often, it is essential to apply a sequential combination of these tools at different stages, such as data conversion and preparation, peak detection from every sample data file, retention time alignment across multiple sample data files, statistical analysis and finally data interpretation to create a fully functional computational pipeline. In this article, we introduce our automated bioinformatics pipeline for metabolomics data analysis, its software architecture, and the user interface that complements the NGMS analytical workflow for application in the diagnostic screening of IEMs. We demonstrate its application in a case study of an IEM diagnosed patient, shedding light on the complete process from sample measurement until interpretation, and how this workflow enables laboratory specialists to perform data processing and analysis in a highly efficient, reproducible, and traceable manner, adhering to ISO:15189 regulations. We also highlight the need for and importance of a robust bioinformatics pipeline design that is reproducible and easy to modify and extend, specifically in the context of use in diagnostics. The focus of this article does not include details on the NGMS analytical approach, data processing algorithms and statistical methods in the context of our workflow; for details on such information, please see [10].

## 2. Pipeline Design and Architecture

The NGMS procedure can be broadly categorized into four steps: data acquisition, quality control, data processing, and data interpretation. This is schematically represented in Figure 1.

High-resolution untargeted metabolomics data are acquired in vendor-specific **.d* file format using an Agilent (Santa Clara, CA, USA) 1290 UHPLC coupled to an Agilent 6540 or 6545 QTOF mass spectrometer. A worklist, in *.*wkl* format, is generated using Agilent MassHunter™ Acquisition (version 10.1 Build 10.1.48) software, and contains information about the analytical run, samples, and datafiles.

Several quality control steps are necessary to ensure the quality of the acquired raw data is up to diagnostic standards. The QC reporting tool assists in this process and is described in detail in Section 4. 

The data processing phase of the NGMS pipeline consists of mass feature alignment, metabolite annotation and statistical selection steps, as displayed in Figure 2. For more details on these processing steps, please see [10]. Three components are responsible for managing the data processing and are used in a sequential order—the storage tool, the workflow starter, and the workflow engine.

### 2.1. Storage Tool

The storage tool transfers the **.d* format raw data files from the computer connected to the mass spectrometer to a read-only storage system. The worklist file is stored together with the measurement data. The tool also converts the raw data files to a vendor-independent **.mzML* generic file format using msConvert [17]. The storage tool also verifies the integrity of the worklist and verifies data transfer using checksums. A screenshot of the Storage tool is displayed in Figure 3a.

### 2.2. Workflow Starter

The workflow starter is an interactive application that allows a user to initiate processing of the acquired data by the workflow engine. The user needs to select the measurement data for positive and/or negative ion mode and specify all parameters required by the workflow. This information is saved in a read-only file in **.json* format for traceability and reproducibility. A screenshot of the Workflow starter displaying the main window with all the existing sessions and a second window with the necessary parameters to start a new session is displayed in Figure 3b.

### 2.3. Workflow Engine

The data processing steps in Figure 2 are executed consecutively by an in-house developed workflow engine on a high-performance computing (HPC) cluster. The principal design objective of the workflow engine is to dispatch job steps securely and efficiently across centrally managed compute and storage resources. The workflow engine also tightly integrates with HPC when managed by a SLURM job scheduler [18].

The workflow engine itself is a containerized suite of tools, designed for both scalability and reproducibility, and allows containerized as well as conventional workflow components to be declared in user-defined job specifications. All software components in the NGMS pipeline are containerized using Singularity [19]. These containers encapsulate both the individual software components as well as their dependencies to ensure portability and reproducibility. All processing data are stored on a read-only storage system.

## 3. Data Interpretation

The interpretation tool enables users to interactively and concurrently browse the mass features detected during the data processing stage. The main purpose of the tool is to allow technicians and laboratory specialists to identify features that are likely clinically relevant for the patient. Technicians will manually validate clinically relevant features in the raw data, to control for false-positive results caused by artifacts during data processing automation. A screenshot of the interpretation tool graphical user interface is displayed in Figure 4, along with the case study in Section 5.

The interpretation tool supports filtering of the detected mass features based on the mass-features-related properties such as retention time, *m*/*z*, *p*-value, fold change, metabolite annotations, etc., determining which features are displayed and how they are ordered. For application in diagnostics, a filter preset is used based on pre-configured parameters and a panel of 340 known diagnostic metabolites—please refer to [10] for the composition of this panel. However, in a research setting, users can opt for different filtering parameters, or no filtering at all, to leverage fully the untargeted metabolomics approach, or ‘open the metabolome’, in analogy to exome panel analysis followed by open exome evaluation. Retention times on our UHPLC-QTOF-MS setup are only known for metabolites in our diagnostic metabolite panel. Therefore, metabolites outside the diagnostic panel are annotated with lower confidence, based solely on *m*/*z*. 

The tool includes the following features for interpretation:A list of measured patient samples available to the current session;A comprehensive per sample feature table;
○Concatenating positive and negative ion mode results;○Including mass spectrometry data, statistical metrics, and links to third party metabolite and pathway databases;○With unmodifiable result columns, to safeguard result integrity; ○Users can add per-feature annotations describing diagnostic relevance based on assigned user roles (i.e., data analyst, clinical laboratory specialist) and the changes are logged in an audit trail; Bar plot-based visualization for comparing the detected sample features against quality control, validation, or other patient samples;A collaborative review and approval process of each sample for patient diagnosis based on configurable user roles; andReal-time updates to each sample’s status as users collaboratively browse, annotate and review features.

## 4. Validation and Quality Control

By setting up extensive validation and quality control protocols, both the analytical and bioinformatic workflows for NGMS have acquired ISO 15189 accreditation through the Dutch Accreditation Council (RvA, M090, 2021). The exact analytical quality control procedure is described in [10]; below, we address the partial automation of the analytical quality control procedure and the validation of the automation itself.

### 4.1. Quality Control of Data

To ensure satisfactory quality of metabolomics data for diagnostic use, it is crucial to incorporate various quality control checks in the NGMS workflow. We have implemented quality control both at the analytical level as well as for the data processing. Through this strategy, both the UHPLC-QTOF-MS performance, as well as the NGMS pipeline performance are thoroughly monitored.

Before the acquisition data is processed by the NGMS pipeline, analytical quality control is performed, the actual compounds that are monitored are elaborately described in [10]. A part of the analytical QC procedure is automated by the QC Reporting tool. The FindByFormula functionality implemented in the Agilent MassHunter Qualitative Analysis software (version 10.0 Build 10.0.10305.0) is applied to the raw data to detect the external standards in the analytical quality control samples, and the internal standards in each patient sample. The exact internal and external standards and their required recovery thresholds are fully described in [10]. MassHunter generates a compound list containing the retrieved internal and external standards. The QC Reporting tool subsequently generates an excel-based report that enables efficient and traceable evaluation of the following analytical quality control aspects:Repeatability of retention time;Repeatability of response;Mass accuracy.

A screenshot of the QC reporting tool is provided in Figure 5.

### 4.2. Validation of Data Processing

For data processing quality control, a sample termed ‘validation plasma’ in which several IEM-related metabolites have been spiked in diagnostic concentrations is processed in identical manner to patient samples. This is a new quality control step that has been added for data processing to the NGMS workflow [10]. During data processing, several plots are generated including a PCA plot, extracted ion chromatograms and a retention time alignment plot. After data processing, the diagnostic output of the validation plasma should include all spiked metabolites as significantly increased. The validation plasma report, containing information on the presence or absence of the spiked metabolites, is automatically generated as an output with every NGMS session. The spiked metabolites included in this report are displayed in Table 1. Only when both analytical as well as data processing quality control have rendered a satisfactory outcome can a diagnostic interpretation session be started.

### 4.3. Validation of Pipeline Releases

To allow for accurate and reproducible patient results, both bioinformatic and clinical validation are performed every time new features are added to the NGMS pipeline and released for use in the form of a major or minor release. Within our software development process, we perform unit, integration, component, and workflow testing which is subsequently reported with every release. For clinical validation, data acquired from a complete analytical run including patient samples is reprocessed with the newer version of the NGMS pipeline and patient results and clinical interpretation thereof are compared to the previous pipeline version. The criterion for a successful clinical validation is that no crucial metabolites should differ which would lead to a false positive or a false negative IEM diagnosis. Additionally, processing of data from the validation plasma samples should always render the spiked metabolites as significantly increased. 

## 5. Case Study: Very Long-Chain acyl-CoA Dehydrogenase Deficiency in an Adult Patient

A 40-year-old male was referred to our adult metabolic clinic because of a history of myopathy, exercise-induced muscle cramps, exercise intolerance, and intermittent rhabdomyolysis. NGMS was performed on a plasma sample as diagnostic screening for a possible inborn error of metabolism. The analytical run generated raw data files (**.d* format) for positive and negative ion mode each being 42.8 GB and 26.7 GB in size, respectively. The NGMS computational pipeline detected 18,028 mass features in the positive mode and 17,567 features in the negative mode. Out of these the mass features found to be altered were 712 in the positive mode and 681 in the negative mode. A total of 42 altered mass features were annotated as metabolites from our diagnostic metabolite panel, based on *m*/*z* and retention time. In all, 595 altered mass features were associated with one or more metabolites present in HMDB based on *m*/*z*. Processing of the data by the pipeline took 1 h 40 min. 

Upon evaluation of results in the interpretation tool, it was immediately apparent that all carnitine ester species of long-chain fatty acids were significantly increased in our patient. This profile is indicative of a diagnosis of late onset very long-chain acyl-CoA dehydrogenase (VLCAD) deficiency (OMIM 201475). VLCAD catalyzes the initial step of mitochondrial β-oxidation of long-chain fatty acids with a chain length of 12 to 20 carbon atoms. Figure 4**.** shows a screenshot of results in the interpretation tool for this patient, in which the feature of tetradecenoyl/C14:1-carnitine, a commonly used VLCADD marker, is highlight ed, which was significantly increased as M + H^+^ adduct with a fold change of ~400. To calculate the fold change, we first take the average of each duplicate measurement and subsequently the median intensity across all other patients in the batch. 

Upon selection of a specific feature, a bar plot is displayed that visualizes the intensity of that feature across all samples measured in the batch as in Figure 6a, where the signal for tetradecenoyl/C14:1-carnitine in the patient is shown in red compared to other individuals in the analytical run in grey, and to the plasma QC pool in blue. Apart from the tetradecenoyl/C14:1, several other metabolites were significantly increased, with fold changes ranging from 20–1200. All the observed perturbed compounds are listed in Table 2 and their corresponding bar plots are displayed in Figure 6b–i. 

These findings were reproduced in a quantitative targeted acylcarnitine assay, and the diagnosis was further confirmed on the enzymatic and genetic level (compound heterozygous variants in *ACADVL*, c.1244C > T p.(Ala415Val) and c.1322G > A p.(Gly441Asp). The patient was referred to a physician specialized in inborn errors of metabolism and was counseled to prevent fasting, avoid ketogenic and high-fat diets, and use medium-chain triglyceride supplementation preceding endurance-type exercise. A Appendix A demonstrating all the steps of the NGMS computational pipeline is included in the Appendix A of this article.

## 6. Discussion

We describe our bioinformatics approach for embedding untargeted metabolomics in the clinical diagnostic process of our metabolic laboratory in an efficient and robust manner. We detail the pipeline software architecture and the first graphical user interface for clinical interpretation of untargeted metabolomics data for screening of IEMs. We also demonstrate the utility of this pipeline by presenting a case study from the clinic. 

In a clinical diagnostic laboratory, all analyses are performed under ISO:15189 accreditation for medical laboratories [20]. This encompasses strict regulations regarding traceability and reproducibility of test results, and thorough validation of analytical processes including software and bioinformatics pipelines. Therefore, for diagnostic application of metabolomics, a reproducible and well-documented data processing and analysis pipeline is crucial, which also incorporates necessary quality checks as well as logging of clinical interpretation of results. Automating the data processing steps in the analysis workflow also reduces manual errors contributing to data integrity. 

We further envision the inclusion of additional modules in the NGMS computational pipeline that can improve diagnostic interpretation and reduce the current time required for clinical interpretation. This includes developing machine-learning-based classifiers to screen patient data for an accurate diagnosis of IEM, without needing a panel of known metabolites, but using the full potential of the untargeted metabolomics data, the so-called ‘open the metabolome’ approach. The classifiers will determine the most discriminant and highly correlated features, characteristic to the condition and would be trained to minimize the number of false-negative and false-positive cases. This can in turn support clinical decision making and help to find novel biomarkers for known IEMs and not yet known IEMs in untargeted metabolomics data. 

Currently, efforts are put into developing a database that includes records on aberrant mass features detected in patients analyzed previously, along with frequently encountered deviations due to medication and other exogenous influences. We suspect that identifying features that are rarely aberrant might help to us find relevant biomarkers, analogous to evaluating the incidence of genetic variants of unknown significance in the general population through databases such as the genome aggregation database (gnomAD) [21]. Including filters based on information from such a mass feature database to the interpretation tool will further automate the interpretation of mass features outside the panel of known IEM-related metabolites. 

In conclusion, we here showcase a robust, reproducible pipeline for application of untargeted metabolomics data in clinical diagnostics of IEM. The software architecture and validation aspects described here can assist other clinical diagnostic laboratories in shaping their design and setup of bioinformatics pipelines.

## Figures and Tables

**Figure 1 metabolites-11-00568-f001:**
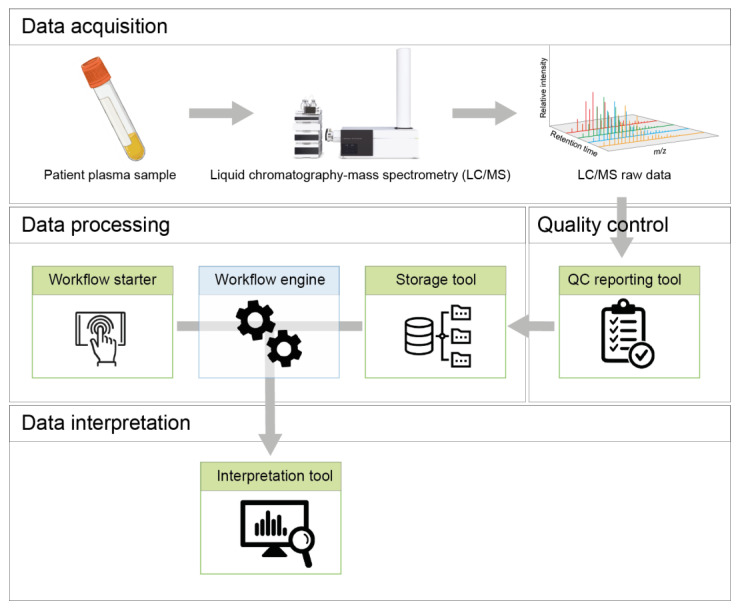
Schematic overview of the NGMS procedure and different tools embedded in the NGMS computational pipeline.

**Figure 2 metabolites-11-00568-f002:**
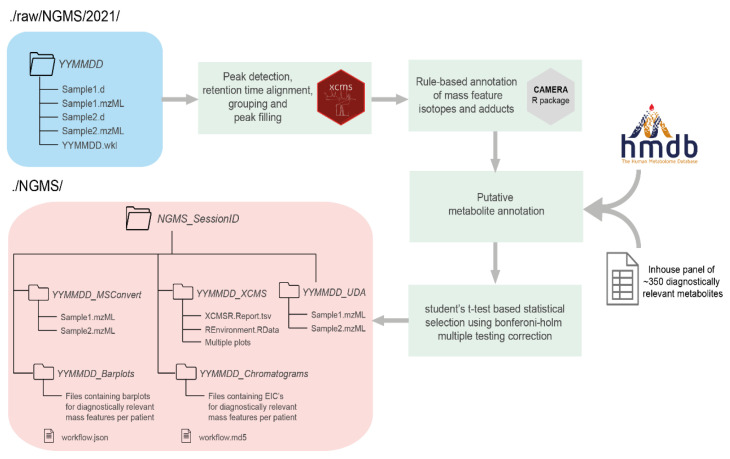
Flow chart of the steps performed by the Workflow engine once a new session is initiated by the user for data processing using the Workflow starter. Input data are colored in blue, the various data processing steps in the pipeline are colored in green, and output data are colored in red. Arrows identify the path of the workflow at each step of the pipeline. There are multiple external tools used in different steps to perform the data preprocessing and annotation like the XCMS R package [14], CAMERA R package [15] and the Human Metabolome Database (HMDB) [16].

**Figure 3 metabolites-11-00568-f003:**
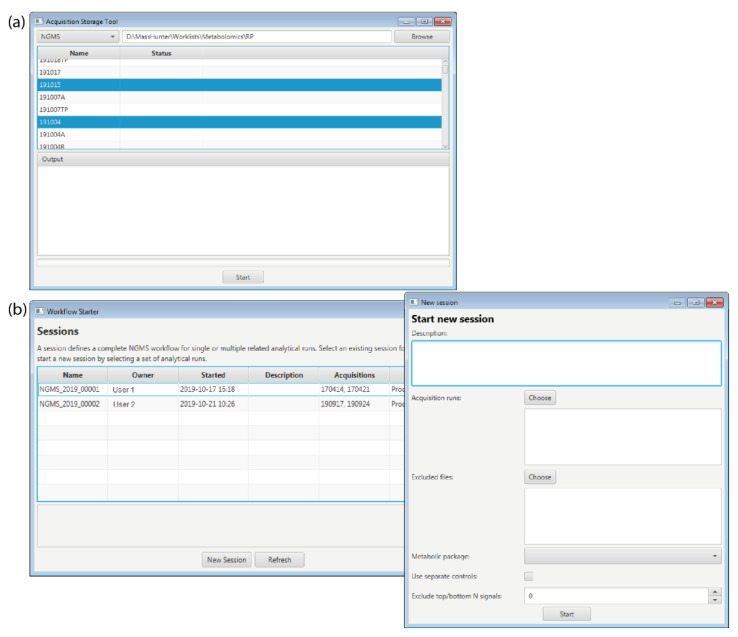
Screenshots of the graphical user interface of the (**a**) Storage tool and the (**b**) Workflow starter.

**Figure 4 metabolites-11-00568-f004:**
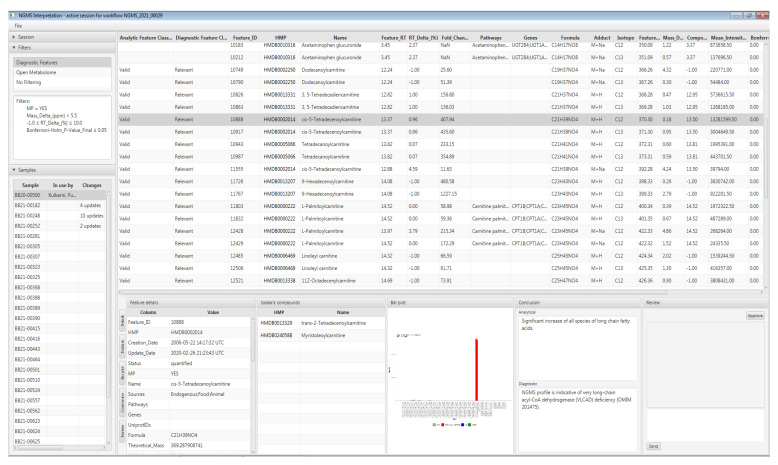
Screenshot of the interpretation tool with the selected patient sample and the diagnostically relevant mass features.

**Figure 5 metabolites-11-00568-f005:**
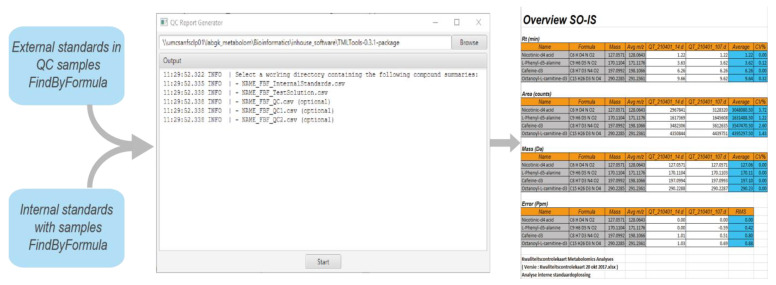
Screenshot of the QC reporting tool along the excel-based report.

**Figure 6 metabolites-11-00568-f006:**
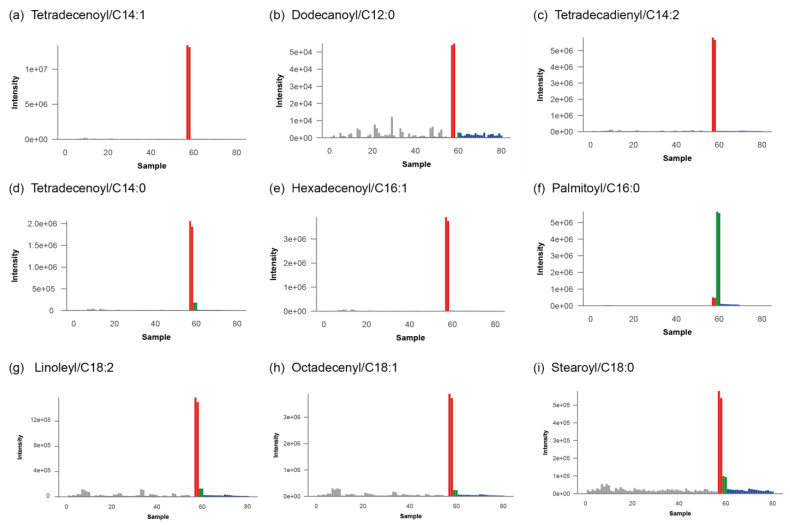
Bar plots for diagnostically relevant mass features corresponding to carnitine ester species of long-chain fatty acids. Patient is shown in red, other patients measured this analytical run in grey, plasma QC pools are shown in blue and validation plasma samples are displayed in green.

**Table 1 metabolites-11-00568-t001:** Table of spiked metabolites as reported for the validation plasma described in Section 4.2.

Name	Feature Mass	Retention Time Delta %	Mean Intensity Patient	ESI−	ESI+
Dihydrouracil	115.0501	4.716981	6,865,418	-	↑
Ornithine	133.0973	0.990099	3,652,369.5	-	↑
Xanthine	153.0407	6.122449	14,056,124.5	-	↑
Ornithine	155.079	0.990099	426,197.5	-	↑
Pimelic acid	161.0808	2.523659	1,699,250.5	-	↑
L-Phenylalanine	167.0892	2.506964	25,365,509	-	↑
Xanthine	175.0227	6.122449	616,580	-	↑
L-Tyrosine	183.0842	3.317535	8,101,318.5	-	↑
Pimelic acid	184.0662	2.523659	1,089,222	-	↑
L-Phenylalanine	188.0678	2.506964	2,532,136.5	-	↑
L-Phenylalanine	189.0713	2.506964	268,255	-	↑
N-Acetylmannosamine	244.0796	1.470588	9,803,499	-	↑
gamma-Glutamylphenylalanine	296.1308	2.763385	20,148,682.5	-	↑
L-Palmitoylcarnitine	400.3423	0	19,211,216.5	-	↑
L-Palmitoylcarnitine	401.3458	0	5,606,300	-	↑
L-Palmitoylcarnitine	422.3247	0	270,923.5	-	↑
Mesaconic acid	129.0194	6.329114	19,645,695.5	↑	-
Xanthine	151.026	3.255814	22,174,103	↑	-
Xanthine	152.0289	3.255814	1,288,583	↑	-
Pimelic acid	159.0661	0.770416	28,452,947	↑	-
Pimelic acid	160.0697	0.770416	7,346,168	↑	-
L-Phenylalanine	164.0714	6.376812	4,407,907	↑	-
L-Phenylalanine	165.0751	2.608696	9,234,293.5	↑	-
L-Tyrosine	180.0664	5.294118	33,613,076	↑	-
L-Tyrosine	180.0661	8.823529	462,782.5	↑	-
N-Acetylmannosamine	256.0597	1.470588	5,667,181.5	↑	-
gamma-Glutamylphenylalanine	293.1141	3.971119	47,200,428	↑	-
gamma-Glutamylphenylalanine	294.1174	3.971119	7,670,664	↑	-

**Table 2 metabolites-11-00568-t002:** List of carnitine esters observed to be perturbed after processing and analyzing the acquired patient sample data using the NGMS computational pipeline.

Bar Plot	Feature Name	*m/z*	Retention Time (min)	Adduct	HMDB ID	Fold Change
(a)	Tetradecenoyl/C14:1	370.295	13.37	M + H	HMDB0002014	407.936
(b)	Dodecanoyl/C12:0	367.264	12.24	M + Na	HMDB0002250	51.387
(c)	Tetradecadienyl/C14:2	368.279	12.82	M + H	HMDB0013331	156.804
(d)	Tetradecenoyl/C14:0	372.311	13.82	M + H	HMDB0005066	354.890
(e)	Hexadecenoyl/C16:1	398.326	14.08	M + H	HMDB0013207	1227.147
(f)	Palmitoyl/C16:0	401.345	14.52	M + H	HMDB0000222	59.362
(g)	Linoleyl/C18:2	424.342	14.32	M + H	HMDB0006469	66.594
(h)	Octadecenyl/C18:1	426.357	14.69	M + H	HMDB0013338	74.586
(i)	Stearoyl/C18:0	428.373	15.08	M + H	HMDB0000848	29.269

## Data Availability

The data presented in this study are available upon reasonable request from the corresponding author. Data are not publicly available due to the terms of the ethical approval.

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
