# Peer review of "Metabolomics-Based Screening of Inborn Errors of Metabolism: Enhancing Clinical Application with a Robust Computational Pipeline"

_metabolites, 2021, doi:10.3390/metabo11090568_

Round 1

Reviewer 1 Report

1) Have the authors incorporated NGMS method into the IS0 15189 system in their lab? Has this method been approved by the national audit institute? This should be mentioned in the paper.

2) The article mentions compatibility with ISO 15189, but this norm requires quite strict rules for validation of the analytical method including accuracy, precision, linearity, sensitivity, repeatability and other parameters. It is not clear from the article how this issue is dealt with. Nor is this clear in reference 10 to which the article refers.

3) It would be useful to indicate if this system is open-source and, if so, to provide a link to download all modules and instructions on how to apply this system in laboratories for IEM diagnosis. 

Reviewer 2 Report

In the present manuscript, Hoegen et al. present the workflow and computational pipeline for  comprehensive, untargeted metabolomic assessment on a large scale. This work is of great interest to the field of metabolic medicine and beyond. By outlining the principles of their data acquisition and interpretation approach, the authors give an overview of their techniques and principles of analysis. Overall, I only have minor comments and would like to congratulate the authors on their work.

Given the fact that their work is by nature a digital one, may I suggest that the authors supply a supplemental file consisting of a video showing the steps of analysis in addition to the screenshot figures throughout the manuscript? This could improve the accessability of their work which is currently somewhat limited by the necessity of descriptions in the manuscript.

The case study chosen for this manuscript offers a nice "hands-on" impression of the use of the diagnostic pipeline. However, the manuscript would benefit either from additional case studies or perhaps from a more diagnostically complex case to underline the power of this novel approach. Ideally, a case that would not have been solved (or only using exceptional resources) using current practice analysis tools.  

Minor comments:

Introduction
l. 53 metabolic profiling (typo)

Pipeline design and architecture

l. 102 "Table 10." - Incomprehensible, please delete.

Reviewer 3 Report

In a continuation of their seminal work in introducing a next generation metabolomics screening platform, the authors elaborate on a data processing pipeline they have developed to semi-automate the deconvolution, processing, feature extraction, analytical quality checks, data processing, data display and interpretation. They discuss a workflow that containerizes vendor software tools, public databases, open source tools, and in-house processes and storage solutions. The work is an important contribution to the development of clinical metabolomics into an efficient and reproducible clinical tool.

Comments:

Line 53: the letter "g" is missing from the word "profiling".

Line 54: would it be clearer if commas were added as such: , called metabolites, ?

Line 102: "Table 10" at the beginning of the sentence appears to be carryover from a poster or previous publication as should be deleted.

Figures 2-5 are too small and should enlarged or saved in an ultrahigh resolution that permits enlargement.

Line 155: Figure 5 is introduced before Figure 4 on line 200. While not a fatal flaw, it is annoying.

The description of the methods make it seem like feature extraction and peak integration are fully software driven. Manual feature curation, confirmation of peak identities, and how is peak integration handled when there are shoulders, overlapping peaks, or isomers of the same m/z are not mentioned.

The analysis of pooled control QC and spiked performance check samples is a nice feature to assessed within batch performance. How many replicates of each type of control is included in each run and if there are replicates, are they randomized within the sequence?

Line 193: This is great work for teaching others. Is there a set recovery threshold that defines "satisfactory recovery"?

Table 1: are these compounds supposed to be the same spike standards defined as the "performance-check solution" in reference 10? Although some are of related chemical classes, they look like entirely different sets of compounds.

In section 5. Validation & release, it is mentions that bioinformatic and clinical validation are performed every time new features are added to the NGMS pipeline including re-running of patient samples. Is there any attempt to batch correct across run days? If so, by what method? If not, what acceptance criteria are used to validate the qualitative comparison between new and old runs?

How many of the features outside of the panel of 340 can you identify? 

Lines 243-244: When you say the M+H+ adduct of C14:1-carnitine was elevated 400-fold, is this the average of the duplicate patient samples, just one of the replicates and is it compared to the mean of C14:1-carnitine in the controls?

Lines 303-305: will any of the software and pipeline features described be available for other clinical diagnostic laboratories to use?

Thank you for bringing this work forward to the scientific and clinical community.
